# Benchmarking Community-Wide Estimates of Growth Potential from Metagenomes Using Codon Usage Statistics

J. L. Weissman,[a] Marie Peras,[b] Tyler P. Barnum,[b] Jed A. Fuhrman[a]

aDepartment of Biological Sciences–Marine and Environmental Biology, University of Southern California, Los Angeles, California, USA
bTrace Genomics, Inc., Redwood City, California, USA

**ABSTRACT** Trait inference from mixed-species assemblages is a central problem in microbial ecology. Frequently, sequencing information from an environment is available, but phenotypic measurements from individual community members are not. With the increasing availability of molecular data for microbial communities, bioinformatic approaches that map metagenome to (meta)phenotype are needed. Recently, we developed a tool, gRodon, that enables the prediction of the maximum growth rate of an organism from genomic data on the basis of codon usage patterns. Our work and that of other groups suggest that such predictors can be applied to mixed-species communities in order to derive estimates of the average community-wide maximum growth rate. Here, we present an improved maximum growth rate predictor designed for metagenomes that corrects a persistent GC bias in the original gRodon model for metagenomic prediction. We benchmark this predictor with simulated metagenomic data sets to show that it has superior performance on mixed-species communities relative to earlier models. We go on to provide guidance on data preprocessing and show that calling genes from assembled contigs rather than directly from reads dramatically improves performance. Finally, we apply our predictor to large-scale metagenomic data sets from marine and human microbiomes to illustrate how community-wide growth prediction can be a powerful approach for hypothesis generation. Altogether, we provide an updated tool with clear guidelines for users about the uses and pitfalls of metagenomic prediction of the average community-wide maximal growth rate.

**IMPORTANCE** Microbes dominate nearly every known habitat, and therefore tools to survey the structure and function of natural microbial communities are much needed. Metagenomics, in which the DNA content of an entire community of organisms is sequenced all at once, allows us to probe the genetic diversity contained in a habitat. Yet, mapping metagenomic information to the actual traits of community members is a difficult and largely unsolved problem. Here, we present and validate a tool that allows users to predict the average maximum growth rate of a microbial community directly from metagenomic data. Maximum growth rate is a fundamental characteristic of microbial species that can give us a great deal of insight into their ecological role, and by applying our community-level predictor to large-scale metagenomic data sets from marine and human-associated microbiomes, we show how community-wide growth prediction can be a powerful approach for hypothesis generation.

**KEYWORDS** codon usage bias, growth rate, metagenomics, microbial ecology

Address correspondence to J. L. Weissman, jakeweis@usc.edu.

The authors declare a conflict of interest. Marie Peras and Tyler P. Barnum are employees of Trace Genomics, Inc.

A fundamental goal of microbial ecology is to characterize the variation of complex microbial traits across time and space and to understand the drivers of that variation (1–4). Many of our best tools for understanding natural microbial communities are molecular and rely on our ability to leverage environmental DNA or RNA sequences to produce insights about community composition (e.g., references 5 and 6). Thus, a major bioinformatic

challenge for microbial ecologists is that of inferring the traits of an organism or organisms on the basis of nucleotide sequences alone (e.g., references 7 to 12).

One trait that has proven to be amenable to such genomic inference is the maximum growth rate of an organism, which can be predicted from various signatures of optimization for rapid translation (13–21). Specifically, organisms capable of fast growth have clear differences in their codon usage patterns relative to organisms which can grow only slowly (13, 21, 22). In particular, because the production of many ribosomes is a universal requirement for rapid growth (23), one can look for codon optimization for rapid translation in the genes encoding ribosomal proteins on a genome as a particularly reliable indicator of maximum growth rate across phyla and even domains of life (13, 24) (and across major metabolic divides; e.g., see Fig. S1 in the supplemental material). Tools have been developed that can predict the maximum growth rate of an organism on the basis of its genome sequence (13, 21), and limited benchmarking on natural and experimentally generated data sets suggests that these methods can also be applied at the community level to metagenomes describing a mixture of organisms to predict a community-wide average maximum growth rate (13, 21, 25). Prediction of the average community-wide maximum growth rate from codon usage statistics is often performed as part of bioinformatics analysis pipelines, to assess how the growth potential of communities varies across environmental gradients (e.g., reference 9). Yet, the performance of these methods on mixed communities has not been well characterized, and it is unclear when we should expect such approaches to work well and/or fail.

Here, we benchmark the performance of our recently developed predictor of maximal growth rate, gRodon (21), on mixed communities of microbes in order to better understand the limitations of this tool. We implement additional bias corrections to the method that greatly improve performance. We go on to provide guidance on data preprocessing and show that calling genes from assembled contigs rather than directly from reads dramatically improves performance. Altogether, we provide an updated tool with clear guidelines for users on when metagenomic prediction of the average community-wide maximal growth rate should be possible. Finally, we apply gRodon to two large-scale metagenomic data sets from habitats of particular interest: (i) the global oceans (26) and (ii) the human body (8, 27). In doing so, we show that community-level maximum growth rate predictions can yield real ecological insights by (i) demonstrating a dramatic decrease in the maximum growth rate of marine communities with depth after 100 m and (ii) demonstrating clear differences in growth potential of the human microbiome across body sites.

## RESULTS AND DISCUSSION

**Implementing a correction for GC content and consistency into metagenome mode.** The original gRodon paper (21) showed that gRodon's default "full" mode for predicting growth rates from individual genomes was not affected by genomic GC content. In contrast, we found that when applied to pairs of concatenated genomes (see Fig. S2 in the supplemental material; simulating two-species communities with even abundances), gRodon v1's "metagenome mode" (MMv1), which applies a simpler codon usage model than "full" mode (21), had a persistent GC bias in which pairs with intermediate GC ranges (~50%) were predicted to be associated with shorter doubling times than pairs with more extreme genomic GC contents (Fig. S2).

At the same time, we found that when predicting the average minimum doubling time of a mixed community of microbes, MMv1 performed best when the organisms in the community shared a common set of preferred codons (Fig. S3). This pattern stems from a key assumption of gRodon that microbes in the same community have codon preferences more similar to those of one another than to those of microbes from different communities. This assumption is based on studies showing that this is indeed the case for many systems (28), where microbes in the same environment tend to have biases toward similar sets of codons in their genomes. It is important that this similarity assumption is met for MMv1 to perform well, because our codon-based growth

predictor compares the bias in codons used by some prespecified set of highly expressed genes (generally taken to be the ribosomal proteins) to a background estimate of codon usage bias (CUB) in the rest of the genes found in a metagenomic sample. If the organisms in a sample do not share a set of codon biases, then the background estimate of bias will be underestimated, leading us to overestimate the codon usage bias of our set of highly expressed genes and underestimate the average doubling time of the community (Fig. S3). We measure the similarity of codon usage across highly expressed genes in a genome or metagenome using "consistency" as a metric (21), which describes how far on average these genes are from each other in terms of codon usage. Because consistency is defined in a somewhat counterintuitive way (higher consistency means more dissimilar codon usage), but also to avoid changing definitions between papers, we represent consistency here as $\Psi$, where larger values of $\Psi$ mean that the ribosomal proteins in a sample have more dissimilar codon usages from one another.

Based in part on developments from an earlier software, growthpred (13), we made three changes to the model used by MMv1: (i) we calculated codon usage bias on a per-gene level, which required an alternative randomization-based approach to calculate the expected codon frequency (see Materials and Methods); (ii) we applied a normalization of codon usage from growthpred that the authors found was helpful in correcting for any GC bias in their predictor (see Materials and Methods); and (iii) we explicitly included GC content as a term in our growth model. Together, these developments effectively corrected the GC and $\Psi$ biases apparent in MMv1 (Fig. S2c and Fig. S3c). In fact, these two biases are closely related, since samples with very skewed GC content also have similar codon usage patterns among their constituents (low $\Psi$), potentially explaining the source of the GC bias in MMv1 (Fig. 1).

On further examination, we found that under certain conditions MMv1 actually outperformed our new bias-corrected metagenome mode (MMBC) on mixed communities (Fig. S4). This occurred when organisms in a sample had very similar codon preferences ($\Psi < 0.6$). Which model performed best appeared to be dependent on a trade-off inherent to the per-gene calculation of CUB in MMBC. When calculating CUB on a single gene, there is a very limited amount of sequence information available with which to calculate the expected background codon frequencies, leading to a high variance in our per-gene estimates of CUB. On the other hand, MMv1 uses the entire metagenome to calculate background codon frequencies, leading to predictions with much lower variance but that were strongly biased for communities where organisms did not share codon preferences ($\Psi > 0.6$). In cases where organisms did share codon preferences ($\Psi < 0.6$), then MMv1 had both low bias and lower variance than MMBC. Since it is straightforward to calculate $\Psi$ for any metagenomic sample, this means that one can decide whether to use MMv1 or MMBC in order to get the best result. We implemented a new mode in the open-source gRodon tool, metagenome mode v2 (MMv2), which automatically switches between MMv1 and MMBC based on a $\Psi$ threshold of 0.6 (Fig. S4).

Finally, we considered the possibility that our MMv2 model was overfit to the training data, which would limit our ability to apply the predictor to new data sets. These models have relatively few parameters overall (Table S1). Yet, overfitting is a particularly salient concern when training models on species data, because species do not represent independent data points (29, 30). In addition to random cross-validation, we implemented a blocked cross-validation approach to control for phylogeny in our error estimates (21, 29). Together, random and blocked cross-validation indicated MMv2 had an equal or better ability to extrapolate across folds than MMv1 and did not appear to be overfit (Fig. S5). Importantly, these single-genome error estimates do not necessarily tell the whole story of a model's ability to predict the average maximum growth rate of a mixed-species community (see below) but rather provide a lower bound for prediction error when applied to such communities.

**gRodon's metagenome mode v2 outperforms other models on synthetic communities.** In order to benchmark gRodon's ability to predict the average maximal growth rates of mixed-species communities, we generated several community-level

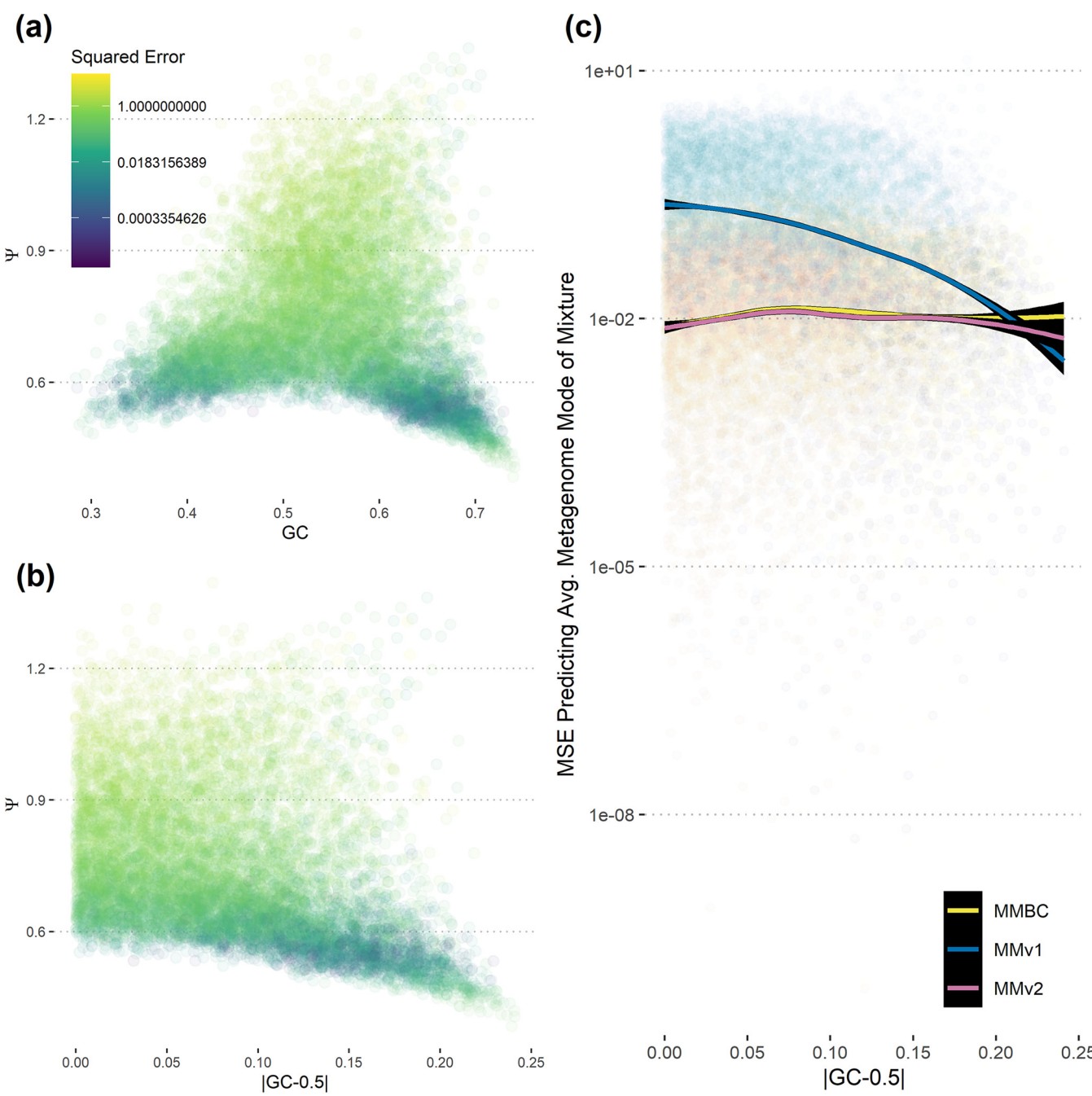

**FIG 1** Samples with highly skewed GC content have low Ψ and low prediction error with MMv1. (a and b) More extreme GC values (far from 0.5) correspond to low Ψ values (<0.6). (c) The graph shows how well each predictor converges to "itself" for various levels of GC skew for genome pairs. That is, if we predict the maximum growth rate of each genome in a mixture using that model and take the average, how well does the prediction on the mixed community estimate that value? Observe that MMv1 performs comparably to (or better than) MMBC and MMv2 only when GC content is highly skewed, corresponding to a low Ψ. MSE, mean squared error.

data sets (Table 1). To start, we generated three sets of 10,000 "genome mixtures" drawn from three sources: (i) a set of RefSeq (31, 32) genomes dereplicated at the genus level (one genome chosen per genus), (ii) a reference catalog of genomes from isolates from the human gut that covers about 70% of these communities (33), (iii) a set of single-cell amplified genomes (SAGs) collected from the ocean surface in an unbiased manner (34). We distinguish "genome mixtures" from true synthetic metagenomes in that our mixtures are simply the concatenated set of coding sequences from the constituent genomes, whereas a "true" synthetic metagenome is produced via the

**TABLE 1** Simulated community-level data used to benchmark gRodon MMv2[a]

| Genome source | Simulation type | Temp simulated | Rel. abund. simulated | No. of simulations | Figure(s)[b] |
|---|---|---|---|---|---|
| RefSeq | Genome mixture | N | N | 1.00E+04 | 1, S2, S3, S4, S5, S8, S9, S17, S18 |
| RefSeq | Genome mixture | Y | N | 1.00E+04 | S10 |
| RefSeq | Genome mixture | N | Y | 1.00E+04 | 2 |
| Marine surface | Genome mixture | N | N | 1.00E+04 | 1, S8, S9, S17, S18 |
| Marine surface | Genome mixture | Y | N | 1.00E+04 | S12 |
| Marine surface | Genome mixture | N | Y | 1.00E+04 | 2 |
| Human gut | Genome mixture | N | N | 1.00E+04 | 1, S8, S9, S17, S18 |
| Human gut | Genome mixture | Y | N | 1.00E+04 | S11 |
| Human gut | Genome mixture | N | Y | 1.00E+04 | 2 |
| Human gut | Synthetic metagenome—reads | N | N | 1.00E+02 | 3, S16, S20 |
| Human gut | Synthetic metagenome—assemblies | N | N | 1.00E+02 | 3, S16, S20 |
| gRodon training data | Genome mixture | N | N | 1.00E+03 | S7 |
| *Mycobacterium* (GTDB207) | Genome mixture | N | N | 1.00E+03 | S19 |
| *Vibrio* (GTDB207) | Genome mixture | N | N | 1.00E+03 | S19 |

[a]Synthetic metagenomes (reads generated from source genomes following a realistic error model) were analyzed both at the read level and after assembly at the contig level as two separate sets of community data. Abbreviations: Y, yes; N, no; Rel. abund., relative abundance.
[b]External links for Fig. S10 through S20 may be found where each of those figures is cited in the text.

generation of synthetic reads from genomic references (we perform such an analysis later on). In this way, we focus the current analysis on issues of prediction on mixed communities only and do not consider possible complications arising from the sequencing or assembly process. Additionally, our concatenation approach allows us to generate very large benchmarking data sets very efficiently (30,000 total generated mixtures). We presently consider mixtures where all organisms are present in equal proportions, though see below for an analysis of mixtures with various species abundances.

In order to benchmark MMv2 against the earlier MMv1 and growthpred predictors, we compared the prediction of each method run on the entire mixture to the average prediction of gRodon's "full mode" on the individual genomes in the mixture. This can be taken as an estimate of the actual error of prediction (since gRodon's full mode is, to our knowledge, the currently best-performing single-genome predictor of maximum growth rate available [21]). Unfortunately, benchmarking against empirically measured growth rates is not currently feasible as the maximum growth rates associated with the genomes used here are either not known or not compiled in any easily accessible way. Any available sets of genome-growth rate pairs have been used to train gRodon and would be inappropriate to use as a benchmark (however, see Fig. S6 for the results of such benchmarking, which also indicate good performance of MMv2).

In general, MMv2 shows improved performance over the earlier MMv1 and growthpred predictors in accurately capturing the average maximal growth rate of a mixture of genomes (Fig. 2a to c). In particular, MMv2 greatly outperforms MMv1 when $\Psi$ is >0.6 (Fig. S7; high $\Psi$ indicates dissimilar codon usage patterns among organisms in a sample). With one exception, MMv2 predictions are not affected by GC content, while MMv1 predictions show a strong dependency on GC across all data sets (Fig. 2d to f). The one notable exception is that in the case of simulated marine communities, MMv2 predicts slower maximal growth for low-GC genome mixtures. This appears to be the product of a real biological pattern, as it is a well-studied phenomenon that slow-growing marine oligotrophs have streamlined genomes with low GC content (possibly as an adaptation to nutrient limitation [35]). MMv2 had a slight performance boost in samples where $\Psi$ was <0.6 due to the bias-variance trade-off between MMv1 and MMBC discussed above (Fig. S8).

All growth prediction models (all gRodon and growthpred modes) similarly implement optimal growth temperature as an optional linear predictor, which may have important impacts on maximum growth rate estimates (13, 21). We also included temperature as an optional predictor for MMv2, and simulations that incorporate

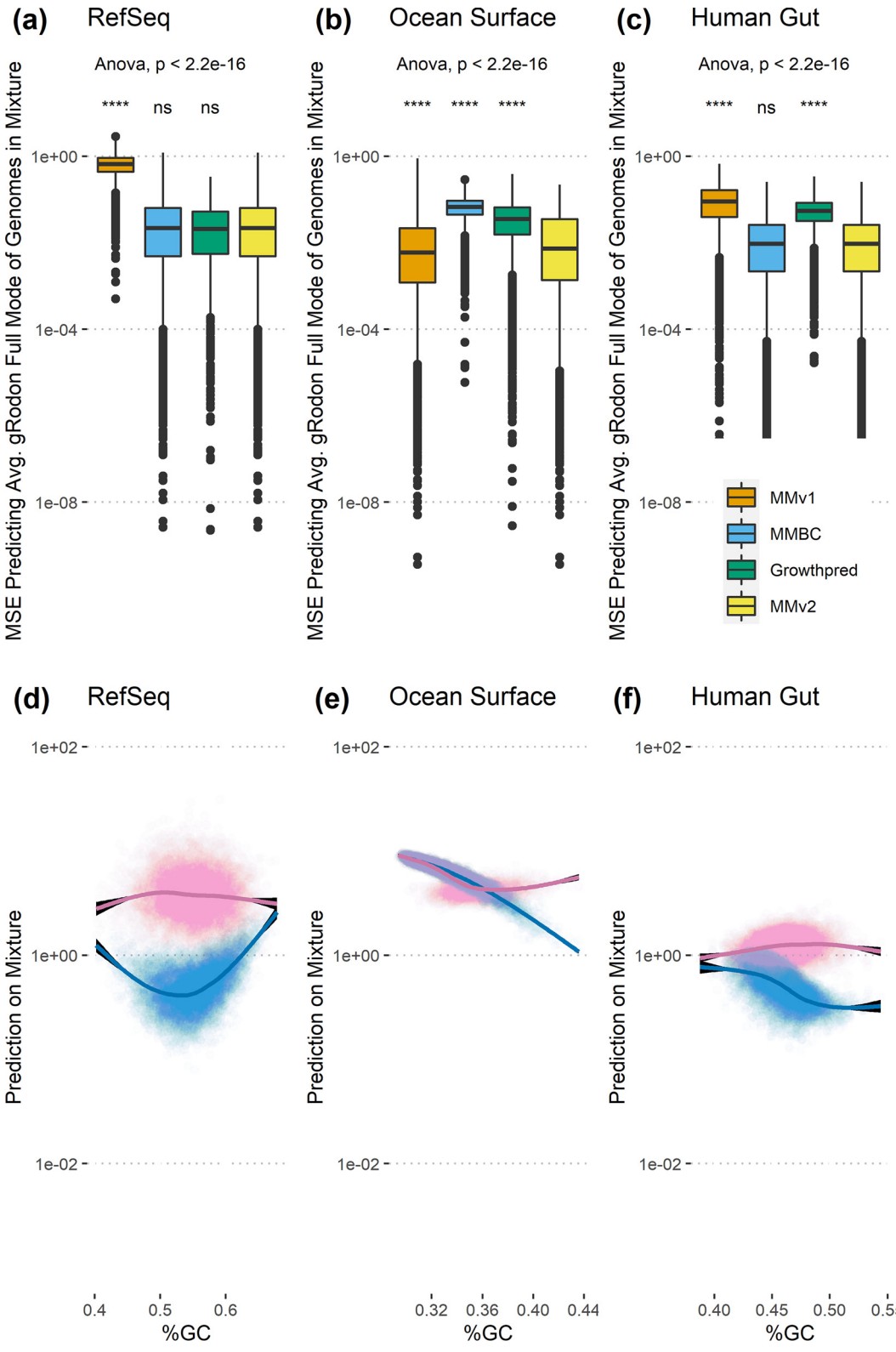

**FIG 2** Metagenome mode v2 (MMv2) outperforms MMv1 and corrects its apparent GC bias. (a to c) For each set of source genomes we calculated the MSE against the prediction for individual genomes using gRodon's "full mode," which gives us the best available baseline for "actual" maximal growth rates for each mixture (21). Analysis of variance (ANOVA) for each sample source shown, with pairwise significance values relative to MMv2 obtained from a Wilcoxon signed-rank test. (d to f) Predicted doubling times from MMv2 (pink) are generally insensitive to the GC content of a sample (see text for discussion of ocean surface mixtures), whereas MMv1's (blue) predictions are strongly affected by GC content. ****, $P \leq 0.0001$; ns, not significant.

temperature variation suggest that including temperature as a variable is important for data sets taken from a wide range of temperature conditions (Fig. S9; see also Fig. S10 at https://doi.org/10.6084/m9.figshare.20440596.v1 and Fig. S11 at https://doi.org/10.6084/m9.figshare.20440356.v1).

Finally, for samples where $\Psi$ is >0.6, MMv2 had longer run times than MMv1, though both methods were able to predict growth rates in a matter of minutes or less (Fig. S12 at https://doi.org/10.6084/m9.figshare.20440383.v1). On a single core on a 2.0-GHz AMD EPYC 7702P processor, on a representative set of metagenomes from the human microbiome (8, 27), MMv2 could predict the average community-wide maximum growth rate in 1.9 min on average, with a maximum run time of 5.9 min for a single metagenome. In contrast, MMv1 could predict the average community-wide maximum growth rate in 14 s on average, with a maximum run time of 52 s for a single metagenome. This difference in run times is a consequence of how MMv2 calculates the background codon usage for each gene individually (see Materials and Methods). Still, even for data sets with thousands of metagenomes, running MMv2 should be feasible in reasonable time frames.

**gRodon's abundance correction improves performance.** One of the key innovations of MMv1 was that gRodon is able to take coverage information into account when calculating the average maximal growth rate of a community in order to effectively weight community members by their relative abundances. In order to determine whether this abundance correction actually improved predictor performance, we again simulated three sets of 10,000 genome mixtures each as described above (simulated from assembled genes of each genome), this time also drawing genome abundances from a lognormal distribution for each mixture which were then assigned to each gene accordingly. We used these simulated abundances in order to calculate the "actual" average maximal growth rate used as a benchmark and then compared our predictions when we either provided gRodon with the relative abundances for each genome or did not. We found that MMv2 had improved predictions when relative abundance information was included (Fig. 3). It appears that improvements were seen primarily in mixtures in which organisms have similar codon usage patterns ($\Psi$ < 0.6; Fig. 3b, d, and f).

**Some assembly is required: gRodon's performance on synthetic metagenomes.** So far, the benchmarking analyses we have described do not account for complications that could arise from the sequencing or assembly process. In order to better simulate a realistic data set, we generated sequencing reads from genome mixtures (36), focusing on organisms found in the human gut (33), in order to generate 100 synthetic gut metagenomes. We then tested MMv2's performance on (i) the genome mixtures used to generate these synthetic metagenomes, (ii) the set of genes called from contigs assembled from the synthetic metagenomes, and (iii) genes called directly from reads from the synthetic metagenomes. For the third analysis, we used gRodon's "fragments" option, which filters out genes shorter than 40 codons long (gRodon typically filters genes shorter than 80 codons long based on guidance from the authors of the CUB statistic used by the program [19, 21, 37]). This short-filter option was necessary to accommodate the short length of the synthetic reads (125 bp) and gives us an idea whether or not it is wise to skip the assembly process altogether for short-read data sets.

We found that MMv2 performed equally well on assembled metagenomes as the original genome mixtures but that performance suffered greatly when MMv2 was applied to genes called directly from reads (Fig. 4). We suspected that this drop in performance was due to the short length of the genes when called directly from reads, which makes it difficult to confidently estimate CUB. Looking at individual genomes, we found that when genes were concatenated to progressively shorter lengths, the variance of their estimated CUB increased greatly, and for very short lengths (typically <120 bp) the CUB was underestimated (Fig. S13 at https://doi.org/10.6084/m9.figshare.20440374.v1). This effect was stronger for ribosomal proteins in fast-growing organisms, including for genes longer than 120 bp (Fig. S14 at https://doi.org/10.6084/m9.figshare.20440395.v1). This suggests that minimum doubling times would be overestimated for genomes or metagenomes with many short genes (i.e., genes inferred from reads), which is consistent with our results on synthetic gut metagenomes (Fig. 4). We also observed that using genes called from reads

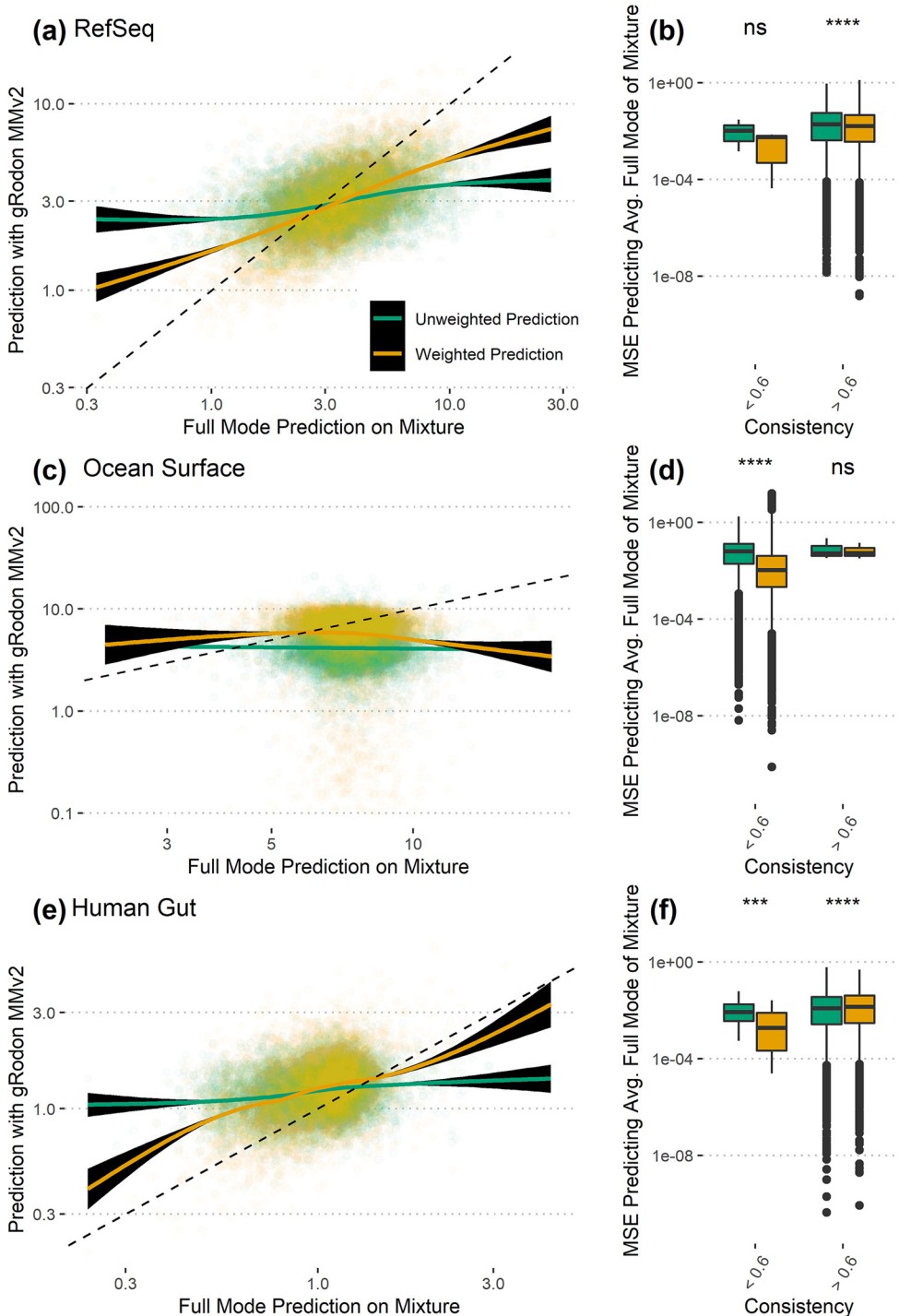

**FIG 3** Abundance-weighted growth predictions with MMv2 are more accurate than predictions that do not account for abundance. (a, c, and e) Doubling time predictions that are weighted for the relative abundance of an organism are more accurate than unweighted predictions. (b, d, and f) The advantages of abundance weighting are seen primarily in mixtures in which organisms have similar codon usage patterns ($\Psi < 0.6$). Pairwise significance values from a Wilcoxon signed-rank test. ***, $P \leq 0.001$; ****, $P \leq 0.0001$; ns, not significant.

from our synthetic metagenomes led to a direct reduction in the estimated CUB of the ribosomal proteins in a sample relative to estimates derived from genes called from assembled contigs (Fig. S15 at https://doi.org/10.6084/m9.figshare.20440389.v1). This analysis suggests that assembly is a necessary step for growth prediction and that short genes

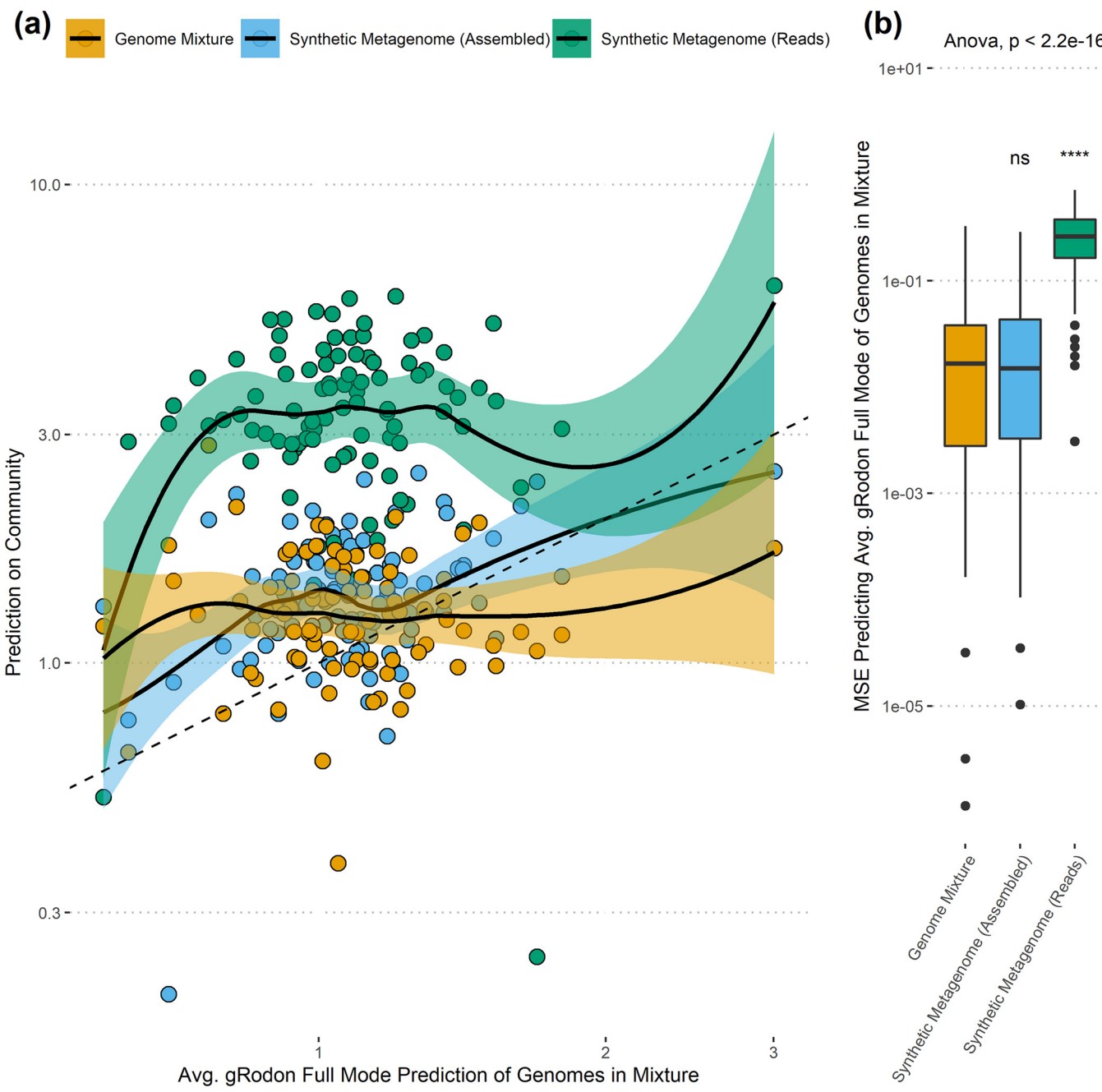

**FIG 4** For short-read data sets, assembly is required for confident maximum growth rate predictions. Doubling time predictions on genome mixtures and synthetic metagenomic assemblies from simulated reads and directly on the simulated reads of synthetic metagenomes for the same sets of organisms were compared. Predictions using genes inferred directly from reads had much higher error rates than predictions on the assembled contigs or genome mixtures, which were not statistically distinguishable. Analysis of variance (ANOVA) across sample types shown, with pairwise significance values relative to genome mixture obtained from a Wilcoxon signed-rank test. ****, $P \leq 0.0001$; ns, not significant.

should be excluded when making such predictions (as done in gRodon's default settings). While growth prediction from short reads (<240 bp) is a desirable feature for future development, it is not recommended using the existing gRodon model.

Interestingly, the assembly process itself did not seem to affect gRodon's prediction performance, with MMv2 performing just as well on synthetic metagenomic assemblies from simulated reads as on mixtures of the original genes used to generate those reads. One might expect that when applied to assemblies, gRodon's performance would suffer marginally due to poor assembly of rare community members, but because those community members contribute so little to the average maximum

growth rate calculation (their contribution being inversely related to their relative abundance), the assembly process seems to have little effect. It is possible that in a hyperdiverse community where even abundant organisms assemble very poorly these results would be different. Nevertheless, by subsampling our genome mixtures, as well as deeply sequenced metagenomes gathered from diverse habitats, we found that even if only a small fraction of the total genes in a community are sampled (around 5% of all genes, and capturing at least 30 to 50 unique ribosomal proteins), MMv2 will provide reliable community-level predictions (Fig. S16 at https://doi.org/10.6084/m9 .figshare.20440398.v1 and Fig. S17 at https://doi.org/10.6084/m9.figshare.20440386.v1; also see unweighted predictions with no correction for relative abundance in Fig. S18 at https://doi.org/10.6084/m9.figshare.20440380.v1 and Fig. S19 at https://doi.org/10 .6084/m9.figshare.20440392.v1). While the actual cutoff for accurate prediction will vary across communities (the 30- to 50-gene cutoff is apparently appropriate for those with a lognormal species abundance distribution; more may be required for more uniform abundance distributions), these simulations suggest that gRodon can be applied to relatively shallow metagenomes.

**Growth prediction on large metagenomic data sets from the global oceans and the human gut.** We ran MMv2 on two large metagenomic data sets to get a sense of how the model would perform on real data: (i) the BioGEOTRACES data set, which consists of 610 globally distributed marine metagenomes (26), and (ii) the Human Microbiome Project (HMP) Illumina WGS Assemblies (HMASM) data set, which consists of 749 metagenomes sampled from a large number of individuals and body sites (8, 27). These data sets describe dramatically different microbial communities living in dramatically different environments, with marine systems typically being dominated by slower-growing organisms living under oligotrophic conditions and the human microbiome generally being composed of much-faster-growing organisms living under nutrient-rich conditions (9, 13, 21).

First, we found that our simulated genome mixtures used for benchmarking did a reasonably good job of recapitulating natural distributions of $\Psi$ (i.e., codon similarity patterns across organisms) from the same environments. That is, the distribution of $\Psi$ seen in our "Marine Surface" mixtures overlapped well with the distribution of $\Psi$ seen in the BioGEOTRACES metagenomes (though the distributions were still statistically distinguishable; Kolmogorov-Smirnov test, $P = 1.17 \times 10^{-7}$ [Fig. 5a]). Similarly, our "Human Gut" mixtures had a $\Psi$ distribution that overlapped the stool samples in the HMASM data set (statistically indistinguishable means; Kolmogorov-Smirnov test, $P = 0.056$ [Fig. 6a]). In both cases, the $\Psi$ distribution of environmental mixtures and actual metagenomes was shifted toward much lower values than those seen in the "RefSeq" mixtures. This implies that environment-specific genome mixtures do a much better job of replicating at least one feature of natural communities than do sets of genomes drawn at random from genomic databases.

Second, we found that MMv2 allowed us to make specific hypotheses about variation in community-wide average maximum growth rates in marine and human systems. Among the BioGEOTRACES samples, we noted a distinct decrease in the community-wide maximum growth rate with depth after 100 m, consistent with predictions from other groups and with decreasing energy inputs to ocean systems with decreasing light (Fig. 5b) (9). Among the HMASM samples, we found differences in the community-wide maximum growth rate across body sites, with the oral microbiome of the tongue and cheek having the fastest maximum growth and the gut and tooth plaque having the slowest (Fig. 6b). These likely have to do with differences in the kinds of substrates available across environments, where more complex carbon sources are broken down in the gut relative to the mouth (e.g., reference 38). Finally, we found clear differences in the average community-wide maximal growth rates across environments (mean BioGEOTRACES doubling time of 11.1 h and mean HMASM doubling time of 1.6 h; Wilcoxon rank sum test, $P < 2.2 \times 10^{-16}$ [Fig. 5c and Fig. 6c]), with 92% of HMASM samples having a higher community-wide maximal growth rate than the highest seen in a BioGEOTRACES sample. Importantly, many of

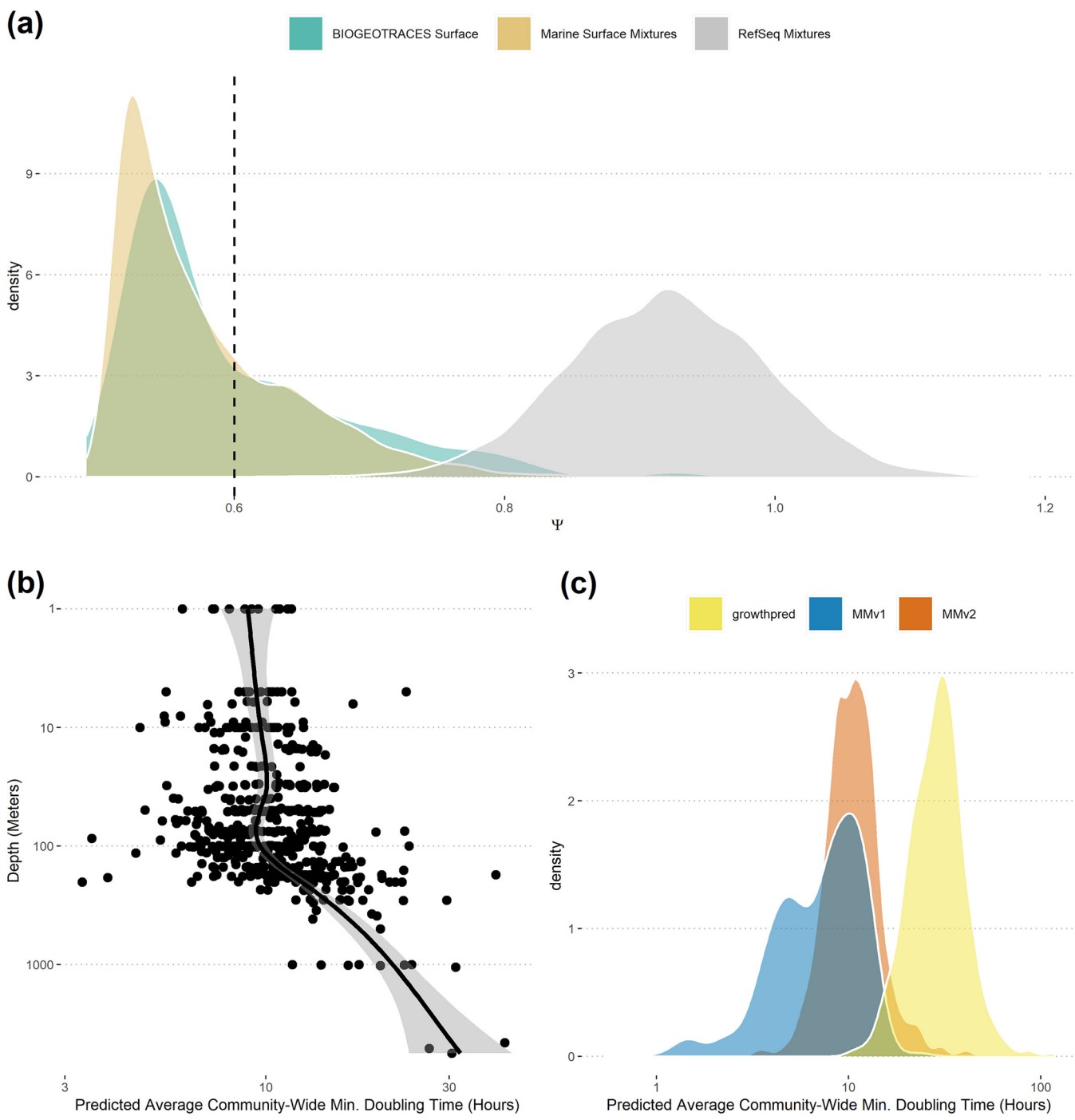

**FIG 5** Natural marine communities as a test case for MMv2. (a) The distribution of Ψ across organisms among BioGEOTRACES samples taken from a depth of <100 m largely overlaps the Ψ distribution from our "Ocean Surface" mixtures, but not our "RefSeq" mixtures. Ψ is a measure of codon usage dissimilarity across highly expressed genes in a sample, and low values of Ψ indicate high similarity (see Results and Discussion). (b) MMv2 applied to the BioGEOTRACES samples recovers a pattern of increasing average minimum doubling time of marine communities with depth after 100 m. (c) Relative to MMv2, MMv1 predictions are strongly biased toward shorter doubling times for BioGEOTRACES samples.

these ecological patterns may have been masked if we applied MMv1, which in general was strongly biased toward higher inferred maximum growth rates (Fig. 5c and Fig. 6c).

**Conclusions.** We developed gRodon MMv2, an improved predictor of the community-wide average maximum growth rate for mixed-species metagenomes, and implemented this predictor in an open-source R package. We provide extensive benchmarking to demonstrate that MMv2 outperforms previous growth models and to provide

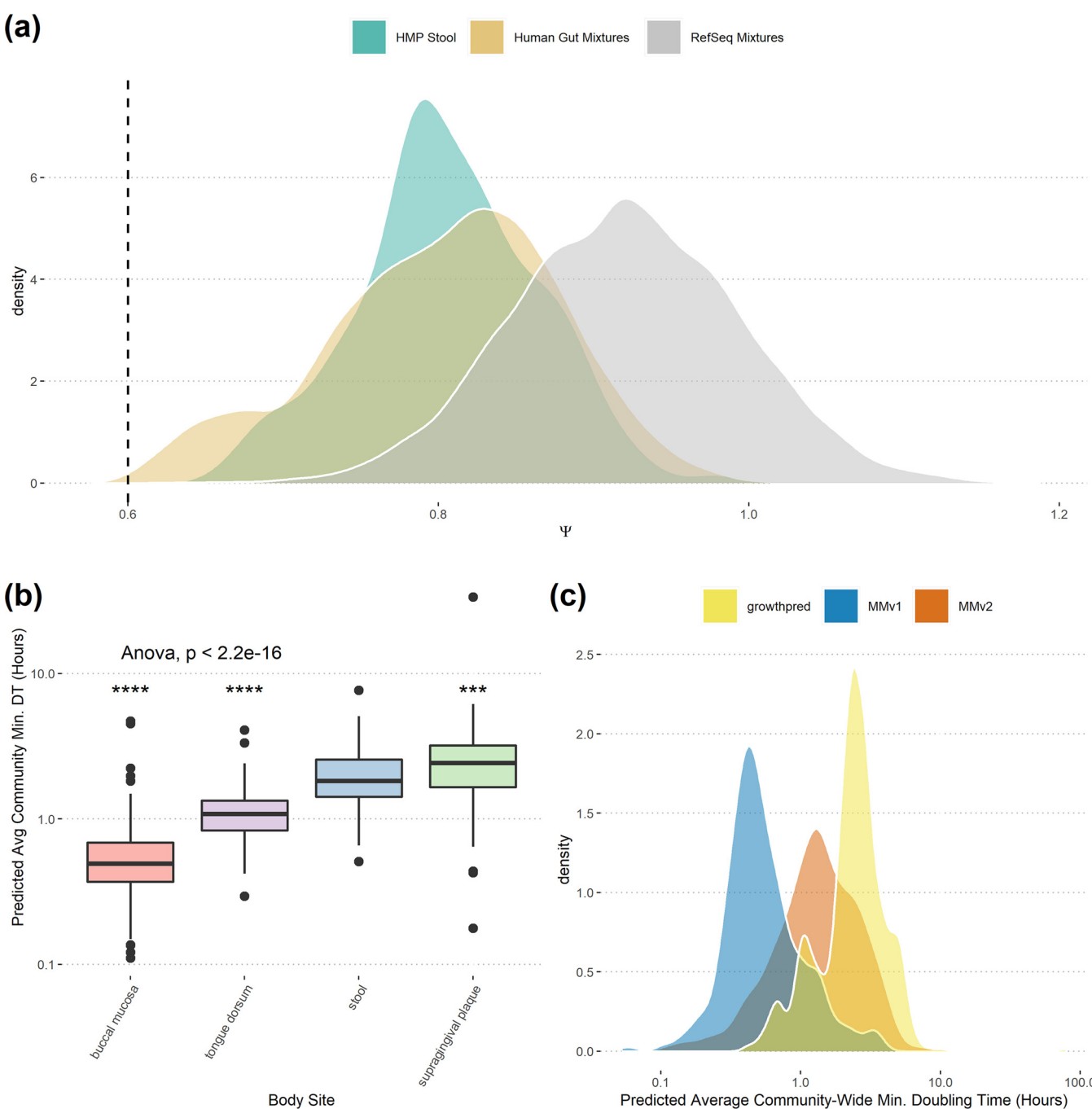

**FIG 6** Natural human-associated communities as a test case for MMv2. (a) The distribution of $\Psi$ across HMASM samples taken from a human stool largely overlaps the $\Psi$ distribution from our "Human Gut" mixtures but not our "RefSeq" mixtures. $\Psi$ is a measure of codon usage dissimilarity across highly expressed genes in a sample, and low values of $\Psi$ indicate high similarity (see Results and Discussion). (b) MMv2 applied to the HMASM samples recovers differences in growth rates across human body sites (sites with >100 samples). Analysis of variance across sites shown, with pairwise significance values relative to stool obtained from a Wilcoxon signed-rank sum test. (c) Relative to MMv2, MMv1 predictions are strongly biased toward shorter doubling times for HMASM samples. ***, $P \leq 0.001$; ****, $P \leq 0.0001$; ns, not significant.

guidance on when our predictor's performance will suffer. For example, we showed that weighting predictions by species abundances improved overall performance and that assembly was a necessary step for reliable prediction on short-read data sets. The development of predictors that work well on unassembled short-read data is an active area of development and would greatly expand the utility of our tool. We emphasize that MMv2 shows improvement only for community-level prediction from mixed-species communities and that for individual genomes users should still use gRodon's "full" mode (21),

which is the best method available for those data types. In the rare case where reference genomes are available for all species present in a metagenome, applying gRodon to the reference database and using coverage to estimate relative abundances in order to calculate the weighted average maximum growth rate, rather than applying metagenome mode directly, is likely to give the best results. Nevertheless, the ability of MMv2 to resolve cross-species differences, even across closely related organisms (Fig. S20 at https://doi.org/10.6084/m9.figshare.20440371.v1), appears to be robust.

By applying MMv2 to large-scale metagenomic data sets, we are able to derive insights about the variation of the community-wide average maximum growth rate across marine and human-associated environments. For example, we saw clear decreases in the community-wide maximum average growth rate with depth, in agreement with other studies (9). At the same time, we saw strong differentiation among body sites in their average maximal growth rates, potentially related to the availability of different growth substrates across sites. Together, these results demonstrate the broad utility of our tool and its potential to enable rapid hypothesis generation from metagenomic data sets.

Finally, we note that our benchmarking approach of comparing single-genome inferences to community-level inferences is inherently limited in that the "ground truth" is an inference rather than a data point in itself (however, see Fig. S6). Nevertheless, we found that community-level predictions can clearly recapitulate individual-level predictions without having to extract individual genomes from the community. As more extensive curated trait data sets become available, alternative benchmarking approaches for mixed-species communities may become feasible, and there is a critical need for such databases.

## MATERIALS AND METHODS

Code for all analyses presented can be found at: https://github.com/jlw-ecoevo/gRodon2-benchmarking. The R package gRodon with MMv2 implemented can be found at https://github.com/jlw-ecoevo/gRodon2.

Sequence handling in R was performed using the Biostrings package (39). Data visualization was performed using ggplot2 and ggpubr (40, 41).

**Data.** Out of the set of RefSeq annotated assemblies (31) incorporated in the original EGGO database (21), we randomly sampled one genome per genus (following the NCBI taxonomy [42]) to generate a set of nonredundant genomes spanning 2,976 genera. These comprised our "RefSeq" genome set. Genomes of human gut isolates were obtained from the Zou et al. (33) set of 1,520 sequenced cultured isolates. Marine SAGs were obtained from the Pachiadaki et al. (34) GORG-Tropics database of 12,715 organisms. All genomes were screened, and genomes with <10 annotated ribosomal proteins were removed.

Genomes for mycobacteria and *Vibrio* species were taken from the set of all GTDB207 representative genomes assigned to these genera (43).

Metabolic oxygen use data for each species in the training data set were obtained from the work of Madin et al. (44).

Raw reads, assemblies, and temperature metadata for the BioGEOTRACES data set were obtained from the work of Biller et al. (26). We then ran EukRep v0.6.6 on these assemblies to classify contigs as eukaryotic or prokaryotic and retained only prokaryotic contigs (using settings -m strict –tie prok [45]). Raw reads and assemblies for the HSASM data set (8) were obtained from https://www.hmpdacc.org/hmp/HMASM/#data, and sample temperature was assumed to be 37°C. Assemblies were annotated with prokka (using options –norrna –notrna –metagenome –centre X –compliant [46]). For reads, adapters and low-quality reads were trimmed using fastp v0.21.0 (47), and then cleaned reads were mapped to inferred genes using bwa mem v0.7.12 (default settings [48]) and coverage was quantified using bamcov v0.1.1 (available at https://github.com/fbreitwieser/bamcov).

**Genome mixtures.** To simulate a genome mixture, we randomly sampled 10 genomes from the relevant set of genomes and concatenated annotated coding sequences from these genomes into a single fasta file for that mixture. This was repeated 10,000 times for each data source. Maximum growth rates were inferred for all source genomes using gRodon's "full" prediction mode to derive ground truth against which to benchmark predictions on the genome mixtures.

To simulate relative abundances, the individual abundance of each genome in a mixture was drawn from a lognormal distribution. These relative abundances were then used to weight the calculation of average community-wide maximum growth rate for the gRodon "full" mode benchmark and were additionally passed as gene relative abundances to gRodon when running MMv2 in abundance-corrected mode.

We generated optimal growth temperatures for genome mixtures using the following two-step procedure. First, for each genome mixture a sample temperature between 0 and 60°C was drawn from a uniform distribution. Then, the simulated optimal growth temperature of each species in the mixture was taken as the sum of this sample-wide value and a draw from a normal distribution with mean zero and standard deviation of 10. These organism-level temperatures were used to predict the individual growth rates of genomes (for the benchmark), and the sample temperature was used to predict the community-level growth rate. This approach was used to account for possible variation in an organism's optimal growth temperature (OGT) relative to the conditions under which it may be found.

Training data mixtures were generated from the genomes matching species in the Madin et al. (44) training set that were used to train gRodon (21).

**Synthetic metagenomes.** Synthetic reads were generated from genome mixtures using inSilicoSeq (36). We (using options –n_reads 100M –model hiseq –coverage lognormal) generated synthetic metagenomes with a Hi-Seq sequencing error model and $100 \times 2$ million 125-bp paired reads. Adapters and low-quality reads were trimmed using fastp v0.21.0, and then either reads were assembled with MEGAHIT v1.2.9 (default parameters [49]) and genes were annotated with prokka (using options –norrna –notrna –metagenome –centre X –compliant [46]) or genes were called directly from reads using FragGeneScanRs (using option –training-file illumina_5 [50, 51]). For assemblies, cleaned reads were mapped to inferred genes using bwa mem v0.7.12 (default settings [48]), and coverage was quantified using bamcov v0.1.1 (available at https://github.com/fbreitwieser/bamcov). We annotated genes called directly from reads as ribosomal proteins using blastn (E value cutoff of $10^{-5}$ and a 99% identity [52]) and the ribosomal protein database from growthpred (13). The sets of genes predicted directly from reads were very large (160 million genes on average), so that we subsampled 1% of genes per sample for gRodon prediction (increasing this to 10% did not change our result; see Fig. S21 at https://doi.org/10.6084/m9.figshare.20440377.v1).

**Subsampled synthetic communities and metagenomes.** Subsampled community-level data were generated via sampling genes from genome mixtures or metagenomes with probabilities proportional to their relative abundances—simulated relative abundances in the case of mixtures, coverage in the case of metagenomes—without replacement.

**Bias-corrected growth model.** See Table S1 in the supplemental material for a list of models used in this paper.

The original gRodon MMv1 model was fit as:

$$\Phi_\lambda(\text{doubling time}) \sim \overline{\text{CUB}}_{\text{HE}} \tag{1}$$

where $\Phi_\lambda$ is the Box-Cox transformation with parameter $\lambda$, fitted using the MASS package (53), and $\overline{\text{CUB}}_{\text{HE}}$ is the median CUB of the highly expressed genes (taken to be the set of ribosomal proteins) relative to the average codon usage patterns across the genome, calculated as the Measure Independent of Length and Composition (MILC) statistic (19) using the coRdon package (37). See the work of Weissman et al. (21) for more discussion of the model fitting process.

The bias-corrected model was fit as:

$$\Phi_\lambda(\text{doubling time}) \sim \frac{\overline{\text{iCUB}}_{\text{All}} - \overline{\text{iCUB}}_{\text{HE}}}{\overline{\text{iCUB}}_{\text{All}}} + |0.5 - \text{GC}| \tag{2}$$

using a normalization approach implemented in the original growthpred software by Vieira-Silva and Rocha (13). GC is the genome- or metagenome-wide GC content. Here, $\overline{\text{iCUB}}$ refers to the average CUB calculated using a per-gene background to estimate the expected codon frequencies rather than the whole genomic background. That is to say, the MILC statistic of a gene is normally calculated by comparing observed codon frequencies to a null expectation estimated from genome-wide frequencies. Here, we calculate the expected codon frequency based on the gene itself, controlling for nucleotide frequencies. This is done by repeatedly shuffling the gene sequence to produce a set of simulated genes on which the expected frequencies are calculated (by default, 100 such "null" genes are simulated for each sequence). Because this process is slow, we sample 100 genes from the genomic background (not in the highly expressed set) for which these per-gene estimates are made to calculate $\overline{\text{iCUB}}_{\text{All}}$. This model was fit using the Madin et al. (44) training set used to train the original gRodon model.

Finally, MMv2 applies either the MMv1 or bias-corrected model based on a $\Psi$ threshold of 0.6, with values lower than 0.6 leading to the application of the MMv1 model and values larger than that threshold leading to the application of the bias-corrected model. $\Psi$ is calculated as the mean CUB of the ribosomal proteins using the set of ribosomal proteins to calculate the background expectation of codon frequencies. This essentially measures how distant the ribosomal proteins are from each other in codon usage space and thus is an indicator of whether or not organisms in a metagenomic sample share the same codon biases (see the work of Weissman et al. [21], where $\Psi$ is defined as "consistency").

For comparisons, growthpred v1.0.8 was run from a docker image from the work of Long et al. (25) (https://hub.docker.com/r/shengwei/growthpred) in metagenome mode (-m) using the same set of ribosomal proteins as those for gRodon in all cases.

## SUPPLEMENTAL MATERIAL

Supplemental material is available online only.

**FIG S1**, TIF file, 2.2 MB.
**FIG S2**, TIF file, 1 MB.
**FIG S3**, TIF file, 1 MB.
**FIG S4**, TIF file, 2.8 MB.
**FIG S5**, TIF file, 1.8 MB.
**FIG S6**, TIF file, 3.1 MB.
**FIG S7**, TIF file, 1.4 MB.
**FIG S8**, TIF file, 1.8 MB.

**FIG S9**, TIF file, 2.8 MB.
**TABLE S1**, PDF file, 0.03 MB.

## ACKNOWLEDGMENTS

J.L.W. was supported by a postdoctoral fellowship in marine microbial ecology from the Simons Foundation (award 653212). We also acknowledge support from Simons Foundation Collaboration on Computational Biogeochemical Modeling of Marine Ecosystems (CBIOMES) grant 549943 (to J.A.F.) and US NSF Division of Ocean Sciences (OCE) grant 1737409 (to J.A.F.).

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
