## [Reviewer comments · mSystems]

Benchmarking community-wide estimates of growth potential from metagenomes using codon usage statistics

JL Weissman, Marie Peras, Tyler Barnum, and Jed Fuhrman

Corresponding Author(s): JL Weissman, University of Southern California

Review Timeline:

Submission Date:

August 8, 2022

Accepted:

September 14, 2022

Editor: Nicola Segata

Reviewer(s): The reviewers have opted to remain anonymous.

Transaction Report:

DOI: <https://doi.org/10.1128/mSystems.00745-22>

September 6, 2022

Dr. JL Weissman
University of Southern California
Department of Marine and Environmental Biology
361 Trousdale Pkwy
Los Angeles, CA 90089

Re: mSystems00745-22 (Benchmarking community-wide estimates of growth potential from metagenomes using codon usage statistics)

Dear Dr. JL Weissman: the reviewers agree that all the points that were raised are now properly addressed. Your manuscript has been accepted, and I am forwarding it to the ASM Journals Department for publication. For your reference, ASM Journals' address is given below. Before it can be scheduled for publication, your manuscript will be checked by the mSystems production staff to make sure that all elements meet the technical requirements for publication. They will contact you if anything needs to be revised before copyediting and production can begin. Otherwise, you will be notified when your proofs are ready to be viewed.

Publication Fees:

If you would like to submit a potential Featured Image, please email a file and a short legend to mSystems@asmusa.org. Please note that we can only consider images that (i) the authors created or own and (ii) have not been previously published. By submitting, you agree that the image can be used under the same terms as the published article. File requirements: square dimensions (4" x 4"), 300 dpi resolution, RGB colorspace, TIF file format.

We recognize that the video files can become quite large, and so to avoid quality loss ASM suggests sending the video file via <https://www.wetransfer.com/>. When you have a final version of the video and the still ready to share, please send it to mSystems staff at mSystems@asmusa.org.

Sincerely,

Nicola Segata

Editor, mSystems

Journals Department
S2 Fig: Accept
S6 Fig: Accept
S9 Fig: Accept
S4 Fig: Accept
S5 Fig: Accept
S8 Fig: Accept
S1 Fig: Accept
S7 Fig: Accept
S1 Table: Accept
S3 Fig: Accept